# Systematic Review and Meta-Analysis: Accuracy of Both Gamma Delta+ Intraepithelial Lymphocytes and Coeliac Lymphogram Evaluated by Flow Cytometry for Coeliac Disease Diagnosis

**DOI:** 10.3390/nu11091992

**Published:** 2019-08-23

**Authors:** Fernando Fernández-Bañares, Ana Carrasco, Albert Martín, Maria Esteve

**Affiliations:** 1Department of Gastroenterology, Hospital Universitari Mutua Terrassa, 08221 Terrassa (Barcelona), Spain; 2Centro de Investigación Biomédica en Red de Enfermedades Hepáticas y Digestivas (CIBERehd), Instituto de Salud Carlos III, 28029 Madrid, Spain

**Keywords:** Celiac disease, intraepithelial lymphocytes, TCRγδ^+^ cells, CD3^−^ cells, celiac lymphogram, review, meta-analysis

## Abstract

It has been suggested that in doubtful cases of coeliac disease, a high CD3^+^ T-cell receptor gamma delta^+^ (TCRγδ^+^) intraepithelial lymphocyte count increases the likelihood of coeliac disease. Aim: To evaluate the diagnostic accuracy of both an isolated increase of TCRγδ^+^ cells and a coeliac lymphogram (increase of TCRγδ^+^
*plus* decrease of CD3^−^ intraepithelial lymphocytes) evaluated by flow cytometry in the diagnosis of coeliac disease. Methods: The literature search was conducted in MEDLINE and EMBASE. The inclusion criteria were: an article that allows for the construction of a 2 × 2 table of true and false positive and true and false negative values. A diagnostic accuracy test meta-analysis was performed. Results: The search provided 49 relevant citations, of which 6 were selected for the analysis, which represented 519 patients and 440 controls. Coeliac lymphogram: The pooled S and Sp were 93% and 98%, without heterogeneity. The area under the SROC curve (AUC) was 0.98 (95% CI, 0.97–0.99). TCRγδ^+^: Pooled S and Sp were both 95%, with significant heterogeneity. The AUC was 0.97 (95% CI, 0.95–0.98). **Conclusions**: Both TCRγδ^+^ count and coeliac lymphogram assessed by flow cytometry in duodenal mucosal samples are associated with a high level of diagnostic accuracy for and against coeliac disease.

## 1. Introduction

Coeliac disease (CD) is an immune-mediated systemic disorder elicited by the ingestion of gluten in genetically susceptible individuals. The pooled global prevalence of CD based on serological test results is 1.4% and based on biopsy results is 0.7% [1]. CD is characterized by the presence of a varied array of gluten-dependent clinical manifestations, CD-specific antibodies, *HLA-DQ2* or *HLA-DQ8* haplotypes, and enteropathy [2,3,4,5]. Generally speaking, CD diagnosis presents no difficulties when the biopsy shows severe villous atrophy and crypt hyperplasia. In clinical practice, however, diagnosis is often less straightforward. Diagnostic difficulties arise especially when biopsy findings are borderline. In such cases, there exists the risk of both under- and over-diagnosis.

Tissue transglutaminase IgA class autoantibodies (anti-tTG2) are the serological markers of choice for the detection of CD [3,4,5]. Serological tests have a high specificity and sensitivity, but can fluctuate in cases with mild intestinal damage and a low gluten intake [6,7]. Moreover, serology seems to have a lower sensitivity and specificity in adults [8].

In addition, the overlap between patients with non-coeliac gluten sensitivity and CD patients with a Marsh type I lesion becomes evident and makes differential diagnosis quite difficult [9,10]. Increasingly, clinicians face the challenge of making a diagnosis of patients who choose to live without gluten, without a previous diagnosis of CD. This is challenging, since both the serology and histology of the small intestine are normalized in patients with CD on a gluten-free diet (GFD). In these circumstances, HLA genotyping is of value, since CD is extremely improbable in patients who are HLA-DQ2/8 negative, though it is insufficient in HLA-DQ2/8 positive patients, since 30–40% of the healthy population are also positive.

Other diagnostic approaches beyond conventional histology and serology have been introduced for the diagnosis of CD [11]. The European Society for Paediatric Gastroenterology, Hepatology and Nutrition (ESPGHAN) and the European Society for the Study of Coeliac Disease (ESsCD) guidelines suggest that in doubtful cases, a high CD3^+^ T-cell receptor gamma delta^+^ (TCRγδ^+^) intraepithelial lymphocyte (IEL) count increases the likelihood of CD diagnosis [4,5]. IELs are increased in the mucosa of untreated coeliac patients. In general, these IELs are CD3^+^αβ^+^ T-cell-receptor-bearing cells. However, 20–30% of CD3^+^ IELS are γδ^+^ T-cell-receptor-bearing cells in CD, which comprise fewer than 10% of the IELs in non-coeliac subjects [12]. TCRγδ^+^ IELs are considered to be highly sensitive and specific for CD, and, furthermore, remain elevated despite a GFD [13,14,15]. Non-T-cell CD3^−^ IELs are the second most abundant IEL subset in healthy mucosa. CD3^−^ IELs comprise heterogeneous phenotypes, of which the functions are not clearly elucidated and which are decreased in untreated coeliac patients [16,17].

The assessment of the density of TCRγδ^+^ IELs is, in general, performed with immunohistochemistry techniques in frozen biopsy samples [13]. This is a user-dependent, laborious technique in well-orientated, high-quality samples, but sampling is often compromised and conclusions may be difficult to draw. The assessment of TCRγδ^+^ IELs by flow cytometry allows for a more accurate quantification. Flow cytometry is a powerful analytical tool for the study of small intestinal immune cells, and, in particular, of IEL cells, and is therefore of value in the diagnosis of CD [18]. Using this technique, an IEL pattern that is typical of CD (coeliac lymphogram) has been defined, consisting of both an increase in TCRγδ^+^ IELs and a decrease in CD3^−^ IELs [18,19].

The aim of the present study was to evaluate the diagnostic accuracy of both an isolated increase of TCRγδ^+^ IELs and a coeliac lymphogram assessed by flow cytometry in duodenal mucosal samples for CD diagnosis by performing a systematic review and meta-analysis of the current literature.

## 2. Methods

### 2.1. Search Strategy and Study Selection

Bibliographical searches were performed in the MEDLINE and EMBASE electronic databases according to the following search strategy: (celiac OR coeliac OR gluten-sensitive*) AND (gamma delta* OR lymphogram OR T-cell receptor* OR intraepithelial lymphocyte* OR flow cytometry). Limits: English and Spanish languages (English abstracts of articles written in other languages were also reviewed); humans; from January 1990 to July 2019. Inclusion criteria were: (1) Any article with people of any sex, age, and race that allowed for the construction of a two-way table, extracting true positive, false positive, true negative, and false negative values for TCRγδ^+^ IEL counting and/or coeliac lymphogram for the diagnosis of CD; (2) TCRγδ^+^ and CD3^−^ IELs assessed by flow cytometry; and (3) A clear description of how the diagnosis of CD had been performed. CD was defined as the presence of a positive coeliac serology (antiendomysium antibody and/or anti-tissue transglutaminase antibody), a compatible duodenal biopsy, and a positive clinical and serological response to a GFD. We excluded articles not fulfilling the inclusion criteria, duplicated articles, letters to the editor, editorials, case series, narrative reviews, and those not related to the object of our study after title and/or abstract reading. Bibliographies of all identified eligible studies were also comprehensively searched for studies not identified using the initial search strategy. Where data were missing from the publication, the first and/or senior author was contacted for further information. According to these criteria, two independent reviewers, reaching a consensus when discrepancies appeared, carried out the identification and selection of the studies (F.F-B. and M.E.). The selection process was documented in line with PRISMA recommendations [20].

### 2.2. Outcome Assessment

#### 2.2.1. Data Extraction 

The name of the first author, year of publication, publication country, number of subjects and controls, definition of CD, flow cytometry methodology, cut-offs of TCRγδ^+^ IEL and CD3^−^ IEL, mean or median TCRγδ^+^ IEL count, mean or median CD3^−^ IEL count, and true positive, false positive, true negative, and false negative values for TCRγδ^+^ IEL and/or coeliac lymphogram were recorded in a standardized fashion.

#### 2.2.2. Study Methodological Quality 

The quality of the studies identified was assessed using the QUADAS-2 (Quality Assessment of Studies of Diagnostic Accuracy included in Systematic Reviews) tool [21], which assesses both the risk of bias and applicability of a study. It comprises four domains: patient selection, index test, reference standard, and flow and timing. Each domain is assessed in terms of risk of bias, and the first three domains are also assessed in terms of concerns regarding applicability. Both reviewers also made this assessment, reaching a consensus when discrepancies appeared.

The methodological quality of the flow cytometry methods was assessed following the MIFlowCyt guidelines [22].

#### 2.2.3. Data Synthesis and Statistical Analysis 

The software STATA 16.0 (Stata Corporation, College Station, TX, USA) was used to perform the meta-analysis, using the “midas” command [23]. A diagnostic accuracy test meta-analysis was performed using a multi-level, bivariate mixed-effects model to synthesize evidence. The pooled sensitivity, specificity, positive likelihood ratio (LRP), negative likelihood ratio (LRN), and a hierarchical summary receiver operator characteristics (SROC) curve were calculated from accuracy data, and the corresponding 95% confidence intervals (CI) were further obtained if necessary. The Cochran-Q method and inconsistency index (*I^2^*) were adopted to investigate and quantify heterogeneity among the studies.

Fagan’s nomogram and the likelihood matrix were used to evaluate the clinical utility of both TCRγδ^+^ IEL and coeliac lymphogram [23].

## 3. Results

### 3.1. Search Results

The search strategy returned 958 citations, of which 49 appeared to be relevant. Full texts were subsequently retrieved for detailed assessment. Forty studies were excluded as they did not meet the inclusion criteria of the systematic review. One additional study was identified in the bibliography search of the eligible studies. Among the 10 eligible articles, there were 6 potentially appropriate articles with insufficient data, for which we tried to contact the authors. In the case of two citations, we successfully contacted the senior author, who provided the number of true positive, false positive, true negative, and false negative values. Finally, four potentially appropriate studies were excluded since important information for the systematic review was lacking in two of them and the authors could not be contacted, while two others were excluded because the diagnosis of CD was not clearly described and generated important doubts for the two reviewers. Thus, a total of six eligible studies, including 519 patients and 440 controls, were selected for the analysis (Figure 1) [14,15,24,25,26,27].

### 3.2. Description of the Included Studies

Table 1 summarizes the six selected cross-sectional studies, all of which were focused on European populations, all except one having been carried out in Spain. One of them studied children and adult populations separately [25], and these were included separately in the analysis. For two studies, only the coeliac lymphogram, and for one study, only TCRγδ^+^ IEL were described; thus, there were five eligible studies for the analysis of coeliac lymphogram (in one of them, separately for children and adults), and four eligible studies for the analysis of TCRγδ^+^ IEL diagnostic accuracy.

Table 2 describes the flow cytometry technique used in the selected studies, showing some differences, mainly in the definition of the CD3^−^ IEL subsets.

### 3.3. Qualitative Analysis of Included Studies

The methodological quality of the included studies was evaluated using the QUADAS-2 tool (Table 3). In two studies, the risk of bias about patient selection was high, since they enrolled non-consecutive patients with known disease and a control group without the condition, which may exaggerate diagnostic accuracy [14,15]. There was also a high risk that the conduct of the index test had introduced a bias in one study [15], since the test threshold was not pre-specified, and a multiple logistic regression was developed to calculate the probability of having CD using both TCRγδ ^+^ and CD3^−^ counts, which could lead to overoptimistic estimates of test performance. In three other studies, the test threshold was not described in the paper but was provided by the authors after contacting them and was the threshold pre-specified for clinical use in their labs [14,25,26]. There were no concerns regarding test applicability.

The methodological quality of the flow cytometry methods is summarized in Appendix A.

### 3.4. Quantitative Analysis of Included Studies

Coeliac lymphogram: As mentioned, a coeliac lymphogram consists of both an increase in TCRγδ^+^ IEL and a decrease in CD3^−^ IEL. Pooled sensitivity and specificity (Figure 2) were 93% (95% CI, 89–96%) and 98% (95% CI, 93–99%), respectively. The SROC curve is described in Figure 3A, being the area under the curve (AUROC) of 0.98 (95%, 0.97–0.99), which implies a high diagnostic accuracy for diagnosing CD. There was no significant heterogeneity.

A summary of LRP and LRN with 95% CIs is described in Appendix A, suggesting that the test is useful for confirmation (when positive) and exclusion (when negative) of CD. Fagan’s nomogram is presented in Figure 4A. Among patients with a pre-test coeliac disease probability of 55%, post-test probabilities were 98% and 8% for positive and negative coeliac lymphogram, respectively.

TCRγδ^+^ IEL: Pooled sensitivity (Figure 5) was 95% (95% CI, 82–99%) with significant heterogeneity (*I^2^* = 95.6%; *p* < 0.001). Pooled specificity (Figure 5) was 95% (95% CI, 91–97%), with heterogeneity (*I^2^* = 62%; *p* = 0.05). The SROC curve is described in Figure 3B, being the AUROC of 0.97 (95%, 0.95–0.98), which implies a high diagnostic accuracy for diagnosing CD.

Appendix A and Figure 4B describe the likelihood ratio matrix and the Fagan’s nomogram, respectively. It is suggested that the test is useful for confirmation (when positive) and exclusion (when negative) of CD. Among patients with a pre-test coeliac disease probability of 55%, post-test probabilities were 96% and 6% for positive and negative TCRγδ^+^, respectively.

### 3.5. TCRγδ^+^ IEL in Diseased Controls with Non-Coeliac Atrophy

Two studies included a control group of patients, all suffering from villous atrophy by any cause other than CD (including malignant immunoproliferative diseases (*n* = 5), olmesartan use (*n* = 5), collagenous sprue (*n* = 2), autoimmune disease-associated enteropathy (*n* = 8), Crohn’s disease (*n* = 1), or idiopathic villous atrophy (*n* = 10)) [24,27]. These patients (*n* = 30), displayed a negative TCRγδ^+^ count in 93.3% (95% CI, 78.6–98.1).

### 3.6. TCRγδ^+^ IEL Evolution after a Gluten-Free Diet

Four of the selected studies described the percentage of TCRγδ^+^ IEL in patients on a GFD [14,15,26,27]. One of them prospectively assessed the evolution of this percentage after 1 year on a GFD [14]. In addition, we added data from our series, published only as an abstract [28], also prospectively evaluating the changes in TCRγδ^+^ IEL after a GFD. Ultimately, 201 patients were evaluated (Table 4). Results demonstrated the persistence of the increased values of this intraepithelial T lymphocyte subset after long-term follow-up on a GFD in both children and adults. Regrettably, it was not possible to perform a meta-analysis of these data.

## 4. Discussion

Recent ESsCD guidelines for CD diagnosis states in the “areas of future research” section that studies are needed to evaluate T-cell flow cytometry and make it widely available for clinical use [5]. The results reported herein suggest that both coeliac lymphogram (an increase in TCRγδ^+^ IEL plus a decrease in CD3^−^ IEL) and the isolated increase in TCRγδ^+^ IEL, assessed by flow cytometry in duodenal specimens, would be appropriate tests for CD diagnosis, since they were associated with an AUROC of 0.97–0.98 and LR values representing strong evidence both in favor of and against CD. Coeliac lymphogram was associated with a higher specificity than the assessment of isolated TCRγδ^+^ cells, whereas an isolated TCRγδ^+^ cell count was associated with a higher sensitivity.

The advantages of the use of flow cytometry to assess TCRγδ^+^ cells are considerable compared to other user-dependent techniques [18]. Results are obtained in an objective, quantitative, reproducible way, allowing for the analysis of a greater number of cells than with immunohistochemistry. The concomitant measurement of CD3^−^ IEL adds specificity to the assay, as has been demonstrated in the present and previous studies [18]. Although the analysis of IEL subpopulations is, in general, not needed for CD diagnosis in seropositive patients, it may have high diagnostic value in doubtful cases that involve differentiating between CD and non-CD atrophy, those in which the mucosal lesion is equivocal [29], when HLA-DQ2.5 and HLA-DQ8 are negative [30], or when serum tTG levels are negative or with low titers (mainly those associated with negative EmA) [31]. The World Gastroenterology Organisation Global Guidelines for CD diagnosis recommend second biopsies to be performed in patients in whom the first biopsies and serological tests have been inconclusive (e.g., seronegative enteropathy) [32], but this strategy implies a second invasive procedure and a considerable delay in the final diagnosis. Finally, it may be useful in HLA-DQ2/8^+^ patients presenting with a grade 1 mucosal lesion since it is a non-specific lesion [10,24,33], which is caused by CD in around 5–15% of cases [34], and serology is positive in only 20–30% of them [35]. In this sense, subjecting a patient to an invasive procedure like gastroscopy with biopsies should involve trying to get as much information as possible from the procedure in order to achieve an accurate diagnosis. Taking an additional biopsy for flow cytometry is an easy procedure and may yield substantial information. Most labs in tertiary and even secondary hospitals have a flow cytometer for diagnostic purposes and analyzing the lymphocyte subpopulations in the duodenal mucosa is an affordable technique.

From the 958 articles selected in this systematic review, only 6 of the 49 references that were classified as potentially relevant were finally included in the meta-analysis. Some of the excluded studies had a high methodological quality, but did not meet the inclusion criteria and, consequently, were not useful for the aim of our meta-analysis, since most of them used immunohistochemistry instead of flow cytometry to assess TCRγδ^+^ cells. Other reasons for excluding studies were reasonable doubts on how CD diagnosis was achieved or on how CD was ruled out in the control group, and the absence of the data required to construct the 2 × 2 tables. The six studies included in the present meta-analysis comprised 959 individuals. Most of them were cross-sectional studies carried out in Spain, where the intraepithelial lymphogram technique was first described and is widely utilized, and has been recently included in the Ministry of Health guidelines for the early diagnosis of CD [36]. Two studies included only pediatric patients, three evaluated a mix of pediatric and adult populations, one of them studied children and adults separately, and one included only adults. Since there were only two studies independently assessing an adult population, a meta-regression analysis could not be performed. In addition, the median age of included patients in the study performed only in adults was 55 years, much higher than in other ones; a TCRγδ+ IEL count resulted in 66% sensitivity for CD diagnosis in that study [27], likely being a reason for the heterogeneity observed in meta-analysis. It is noteworthy that the CD patients with normal TCRγδ^+^ IEL in that study were significantly older than those with abnormal values [27]. The authors argued that advanced age may be a factor preventing the characteristic increase of TCRγδ^+^ IEL in CD for unknown reasons, and thus might explain the lower sensitivity found in that study. In the validation cohort of the cut-off for TCRγδ^+^ established in our laboratory, we did not find differences in TCRγδ^+^ sensitivity (≈94%) for CD diagnosis between different age groups, but the patients included were under 50 years of age [37]. It is possible that some subjects with suspected CD who are over 50 years of age could have a negative coeliac lymphogram; thus, further studies will be required to assess the TCRγδ^+^ sensitivity in these patients. Specificity, in contrast, was very high regardless of age.

As mentioned, it has been suggested that TCRγδ^+^ values remain elevated in CD patients despite a GFD. The present systematic analysis included 200 patients supporting this observation. The persistence of the increased TCRγδ^+^ cell values after a long-term GFD (1 year to a mean of 5 years) opens up the possibility of using this biomarker to confirm CD in patients who were started on a GFD, and for whom serology and histology may have yielded misleading results. In addition, it may prove useful for distinguishing between CD and non-coeliac gluten sensitivity in patients who are symptom-free after a GFD and reluctant to undergo a gluten challenge. Further studies are required to evaluate the persistence of the increased TCRγδ^+^ cell populations after longer follow-up.

There were small differences in the flow cytometry technique used between the six studies. Aspects of the sampling, the number of biopsies used, and the sampling locations differed between studies or were not appropriately described. However, unlike histopathological analyses, the IEL pattern is uniformly distributed in the distal duodenum and the duodenal bulb in healthy donors and coeliac patients [38], and the sampling location is therefore not a critical factor. All of the six studies used calcium chelation and shaking for the isolation of IEL, with minor differences between them (time, temperature, and buffer). The number of isolated IELs, the viability of the obtained cell suspension and the amount of starting material were not always reported, which is not critical due to the high efficiency of the isolation protocols [18]. The major differences between the six papers were in the staining panel, gating strategy, and IEL subset definition. Only one of the six papers excluded dead cells in its gating strategy, which is an essential step for reducing nonspecific stained cells that would interfere in the results. Of particular importance is the definition of the CD3^−^ subset, given the heterogeneous nature of this subset and its importance in defining the “coeliac lymphogram”: a consensus regarding the most adequate definition should be reached to reduce confusion in the field. In fact, the CD3^−^ subset evaluated by Nijeboer et al. was completely different from that in the rest of studies (Table 2); it corresponded to a small proportion of IELs lacking surface CD3 but expressing intracellular CD3, which was massively expanded in refractory CD type II [39]. The aforementioned differences in analysis strategy might explain, at least in part, the differences in the cut-off, which might not be transferable from lab to lab. In fact, four different cut-off values for TCRγδ^+^ were used in the six studies. All these differences could account for the heterogeneity observed in the present meta-analysis. Heterogeneity is to be expected in meta-analyses of diagnostic test accuracy, and for this reason, a random effects model was used for the present analysis.

None of the papers fulfilled the criteria to be considered reproducible according to the MIFlowCyt guidelines [22]. Considering the proven clinical usefulness of the IEL pattern, an effort should be made by all researchers to report methodologies at a sufficient level of detail to allow other groups to implement them in a standardized manner.

A limitation of the present study was the heterogeneity of the control group (a small number of patients with non-coeliac atrophy or other enteropathies, and a majority with functional bowel disease patients, gastro-esophageal reflux disease, *Helicobacter pylori* gastritis, parasitic infections, etc., all with normal small bowel histology). In this sense, the absence of a subgroup of seronegative CD among cases and a subgroup with enough non-CD atrophy patients among controls was a drawback, since these are the patient subgroups in which coeliac lymphogram has the highest diagnostic interest. However, two of the included studies provided data on a small sample of non-coeliac villous atrophy patients, suggesting that TCRγδ^+^ count may be useful to rule out CD in this setting. Finally, since all included articles except one were carried out in Spain, it could be argued whether the results might be extrapolated to other populations. However, the increase of TCRγδ^+^ in patients with CD has been described in studies from other countries around the world [13,40,41,42,43,44].

## 5. Conclusions

In conclusion, both TCRγδ^+^ count and coeliac lymphogram assessed by flow cytometry in duodenal mucosal samples have been associated with a high level of diagnostic accuracy for and against coeliac disease. Further studies are warranted to confirm the diagnostic value of the technique in cases in which diagnosis is not straightforward.

## Figures and Tables

**Figure 1 nutrients-11-01992-f001:**
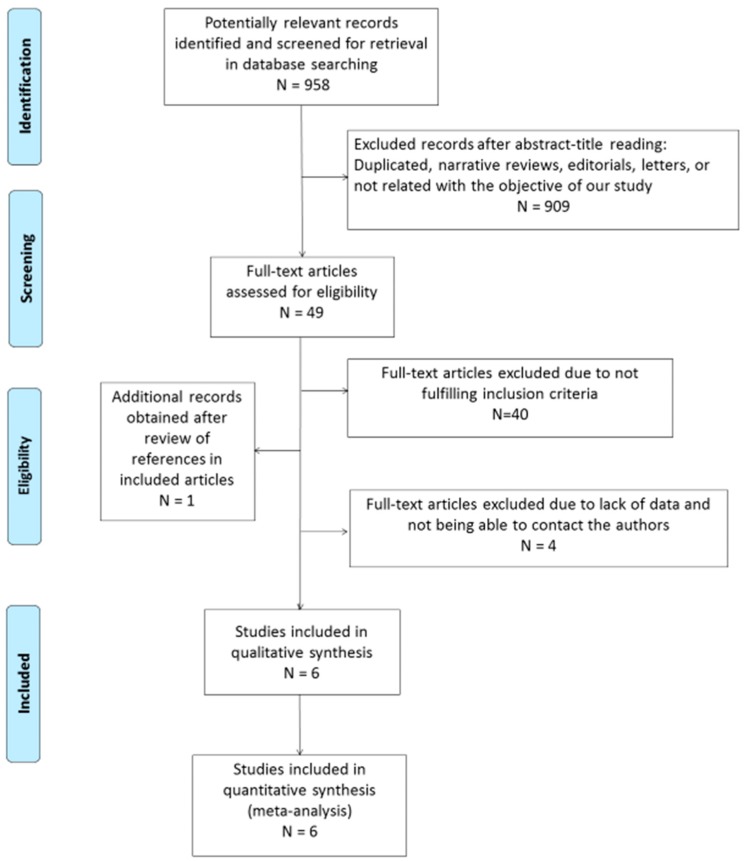
Flow diagram describing the study selection process.

**Figure 2 nutrients-11-01992-f002:**
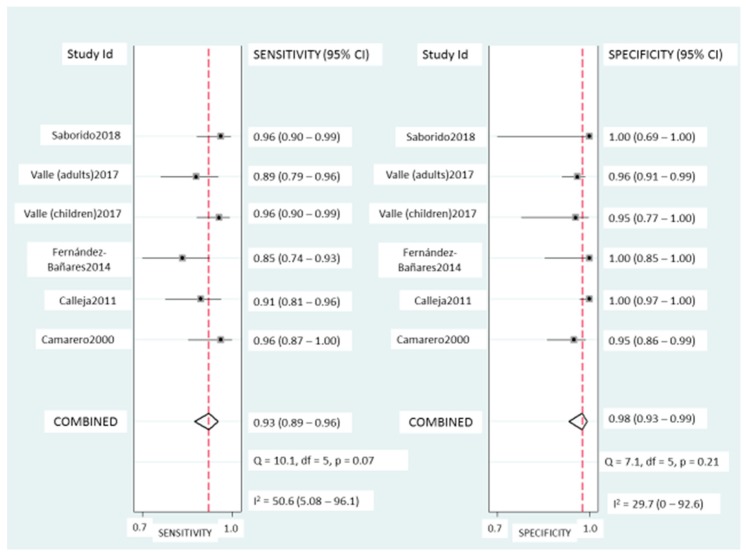
Forest plots showing the pooled sensitivity and specificity of coeliac lymphogram (defined as an increase in TCRγδ^+^ IEL plus a decrease in CD3^−^ IEL) for coeliac disease diagnos.

**Figure 3 nutrients-11-01992-f003:**
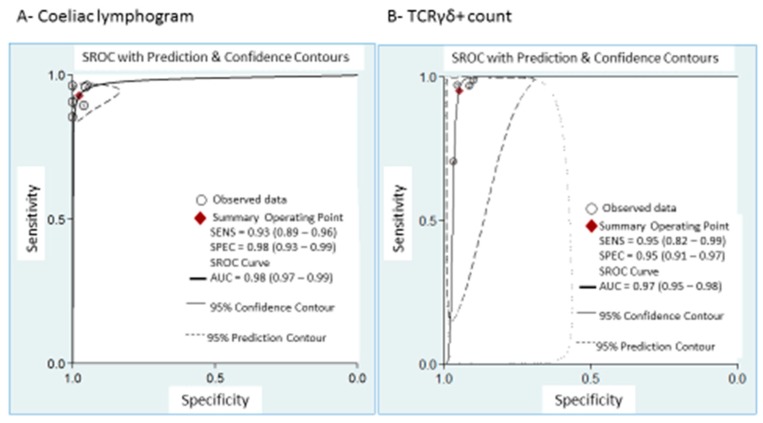
Summary ROC curve (SROC) with confidence and prediction regions around mean operating sensitivity and specificity point of: (**A**) coeliac lymphogram (increase in TCRγδ^+^ IEL plus decrease in CD3^–^ IEL); (**B**) TCRγδ^+^ count.

**Figure 4 nutrients-11-01992-f004:**
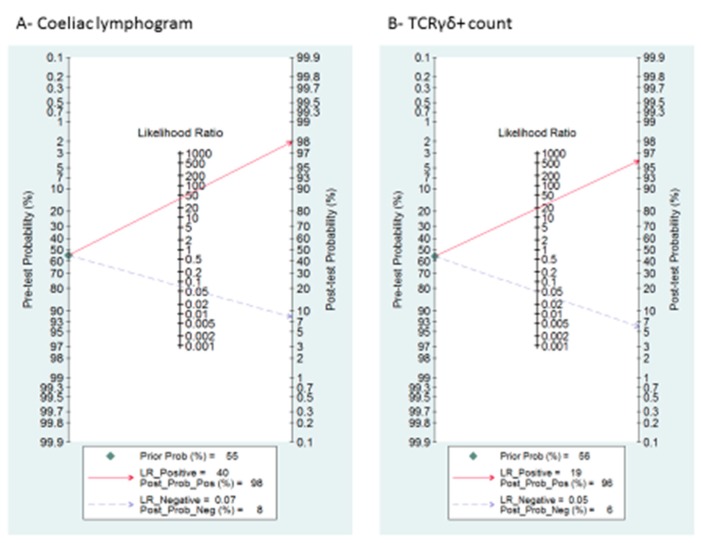
Fagan’s nomogram for the elucidation of post-test probabilities. With a pre-test probability of CD of 55%, the post-test probabilities of CD, given positive and negative index test, were: (**A**) coeliac lymphogram: 98% and 8%, respectively; and (**B**) TCRγδ^+^: 96% and 6%, respectively.

**Figure 5 nutrients-11-01992-f005:**
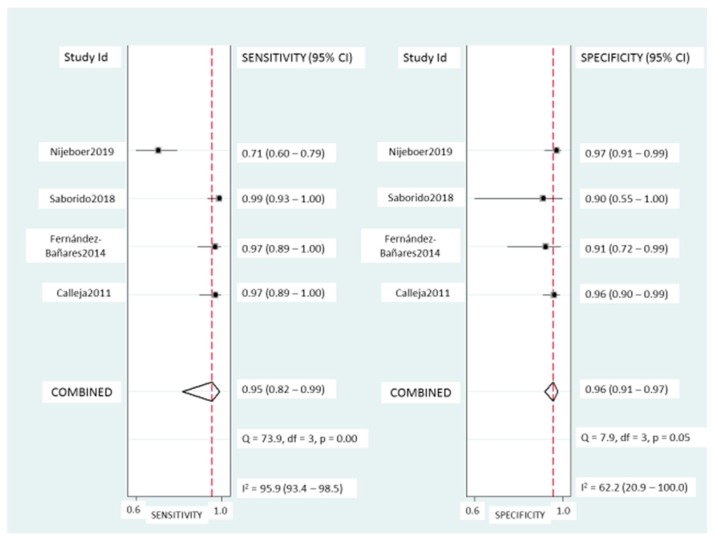
Forest plots showing the pooled sensitivity and specificity of the TCRγδ^+^ count for coeliac disease diagnosis.

**Table 1 nutrients-11-01992-t001:** Summary of selected studies

Study ID	Patients	Controls	Age	γδ^+^ IEL (%)	CD3^−^ IEL (%)	Accuracy of Increased TCR γδ^+^	Accuracy of Coeliac Lymphogram	N. of Subjects with Abnormal Tests
Camarero 2000 [15]	40 CD plus 14 CD on GFD	59 non-CD (dyspepsia, diarrhea, *H. pylori* gastritis) (55 normal histology and 4 enteropathy)	0–18 y	CD Patients: 28 ± 13% (SD) Controls: Mean, 8%; median 5.5% (P10–90: 2.4–21%) P < 0.01	CD patients: 4.9 ± 7.9% (SD) Controls: Mean, 42%; median, 47% (14.4–67.3%) P < 0.01	Not available (N.A.)	S, 94.4%; Sp, 95% (using an equation derived from a logistic regression analysis)	CD lymphogram: 51/54 CD 3/59 controls
Calleja 2011 [14]	66 CD	112 non-CD (dyspepsia, gastroesophageal reflux disease, iron deficiency anemia) with normal histology and negative serology	Children (median age, 5 y, 1 to 14) and adults (median age, 42 y, 15 to 73)	CD patients: 29 ± 1.4% (SEM) Controls: 5.3 ± 0.4% (SEM) P < 0.001	CD patients: 5.6 ± 1.4% (SEM) Controls: 23.1 ± 1% (SEM) P < 0.001	S, 97%; Sp, 95.5%	S, 88%; Sp, 100%	Increase in γδ^+^: 64/66 CD 5/112 controls CD lymphogram: 58/66 CD 0/112 controls
Fernández-Bañares 2014 [24]	50 CD plus 12 potential CD	23 non-CD (8 non-CD atrophy plus 15 *H. p*y*lori* lymphocytic enteritis)	Children and adults (mean age, 29 y)	CD patients: Median, 23% (P25–75: 19–33) Potential CD: Median, 34% (20–37%) Controls: Median, 5% (4–7%) P < 0.001	CD patients: Median, 4% (2–6%) Potential CD: Median, 6% (4–8%) Controls: Median, 22% (15–30%) P < 0.001	S, 97%; Sp, 91%	S, 85%; Sp, 100%	Increase in γδ^+^: 60/62 CD 2/23 controls CD lymphogram: 53/62 CD 0/23 controls
Valle 2017 [25]	161 CD	147 non-CD (negative serology, no atrophy)	95 children (median age 7 y, 0 to 13) and 66 adults (median age 34 y, 14 to 74)	CD patients: Median, 27% (P25–75: 2–62%) Controls: Median 5%, (0–22%) P < 0.001	CD patients: Median, 2% (0–8%) Controls: Median, 20% (1–90%) P < 0.001	NA	Children: S, 96%; Sp, 95% Adults: S, 89%; Sp, 96%	CD lymphogram: Children: 91/95 CD; 1/22 controls Adults: 59/66 CD; 5/125 controls
Saborido 2018 [26]	81 CD	10 non-CD (symptoms of CD, antigliadin antibodies (AGA)+, normal histology, negativization of AGA and symptom resolution on follow-up)	Children; median age, 5 y (1 to 16)	CD patients: Mean, 32.9 ± 13.2% (SD) Controls: Mean, 7.5 ± 9.8% P < 0.001	CD patients: 3.7 ± 8.8% (SD) Controls: 42.4 ± 17.6% P < 0.001	S, 99%; Sp, 90%	S, 96%; Sp, 100%	Increase in γδ^+^: 80/81 CD 1/10 controls CD lymphogram: 78/81 CD 0/10 controls
Nijeboer 2019 [27]	95 CD plus 118 CD on GFD	89 non-CD (symptoms of CD, negative serology and normal histology)	Adults; median age, 53 y (14 to 81)	CD patients: Median, 18.5% (range, 1–58) Controls: Median, 6% (1–15) *P* < 0.001	N.A.	S, 66.3%; Sp, 96.6%	N.A.	Increase in γδ^+^: 67/95 CD 3/89 controls

**Table 2 nutrients-11-01992-t002:** Summary of flow cytometry technique characteristics in selected studies

Study ID	Sample	Treatment for IEL Isolation	Gating Strategy	TCRγδ+ Definition	CD3^−^ Definition
Camarero 2000 [15]	Duodenum or proximal jejunum	RPMI 10% FBS, 1 mM DTT, 1mM EDTAShacker, 60 min, RT	IEL: CD45^+^, lowSSC	CD45^+^TCRγδ^+^	CD45^+^CD3^−^CD7^+^
Calleja 2011 [14]	3 biopsies distal duodenum	RPMI, 1 mM DTT, 1 mM EDTA60 min	IEL: CD45^+^, lowSSC, CD103^+^	CD45^+^TCRγδ^+^ CD103^+^	CD45^+^CD3^−^CD103^+^
Fernández-Bañares 2014 [24]	1 biopsy 2nd part of duodenum	HBSS, 1mM DTT, 1 mM EDTAShacker, 90 min, RT	IEL: CD45^+^, lowSSC	CD45^+^TCRγδ^+^	CD45^+^CD3^−^
Valle 2017 [25]	2 biopsies 2nd part of duodenum	RPMI 10% FBS, 1 mM DTT, 1mM EDTAShacker, 60 min, RT	IEL: CD45^+^, lowSSC	CD45^+^TCRγδ^+^	CD45^+^CD3^−^CD103^+^
Saborido 2018 [26]	1 biopsy	RPMI 10% FBS, 1 mM DTT, 1mM EDTA60 min	IEL: CD45^+^, lowSSC, CD103^+^	CD45^+^TCRγδ^+^ CD103^+^	CD45^+^CD3^−^CD103^+^
Nijeboer 2019 [27]	6 biopsies	PBS, 1 mM DTT, 1 mM EDTA60 min	IEL: CD45^+^, lowSSC	CD45^+^TCR γδ ^+^	CD45^+^surfaceCD3^−^intrCD3^+^ CD7^+^

DTT: ditiotreitol; EDTA: ethylenediaminetetraacetic acid; RPMI: Roswell Park Memorial Institute medium; FBS: phosphate-buffered saline; HBSS: Hanks′ balanced salt solution; SSC: side scatter; IEL: intraepithelial lymphocytes.

**Table 3 nutrients-11-01992-t003:** Results of Quality Assessment of Studies of Diagnostic Accuracy included in Systematic Reviews (QUADAS-2) checklist.

Study	Risk of Bias	Concerns Regarding Applicability
Patient Selection	Index Test	Reference Standard	Flow and Timing	Patient Selection	Index Test	Reference Standard
Camarero 2000 [15]	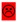	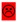					
Calleja 2011 [14]	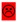						
Fernández-Bañares 2014 [24]							
Valle 2017 [25]							
Saborido 2018 [26]							
Nijeboer 2019 [27]							


 Low risk; 
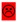
 High risk.

**Table 4 nutrients-11-01992-t004:** TCRγδ^+^ cells in CD patients following a long-term GFD.

Study	Sample Size	Time on a GFD	Mean Age (Years)	TCRγδ^+^ Cut-Off for CD	Baseline TCRγδ^+^ IEL (%)	After-GFD TCRγδ^+^ IEL (%)
Calleja [14]	21	1 year	Adults	>12%	Mean (SEM), 24.9 (3.3) (100% Marsh 3)	Mean (SEM), 25.7 (3.0) (14% Marsh 3a)
Saborido [26]	30	5.4 ± 1.6 years	Mean 10.3 (range, 6–18)	>10%		Mean (SD), 35.9 (16.4); Median, 36.5 (IQR, 25–75) (0% Marsh 3)
Camarero [15]	14	>2 years	Mean 7 (range, 3–14)	>10%		Median, 22 (IQR, 19–36) (0% Marsh 3)
Rosinach [28]	18	>1 year	Mean 25 (range, 5–65)	>8.5%	Mean (SEM), 25.1 (2.4) (100% Marsh 3)	Mean (SEM), 28.2 (2.6) (16% Marsh 3)
Nijeboer [27]	118	NA	Median 53 (range, 12–79)	>13%		Median, 19 (0% Marsh 3)

NA: Not available GFD: Gluten-free diet.

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
