# Peer review of "Systematic Review and Meta-Analysis: Accuracy of Both Gamma Delta+ Intraepithelial Lymphocytes and Coeliac Lymphogram Evaluated by Flow Cytometry for Coeliac Disease Diagnosis"

_nutrients, 2019, doi:10.3390/nu11091992_

Round 1

Reviewer 1 Report

This is a meta-analysis and systematic review about the use of CD3+ gamma delta intraepithelial lymphocytes (IEL) for the diagnosis of celiac disease (CeD). Although serological markers are available, it has been suggested that CD3+ T-cell receptor gamma delta+ intraepithelial lymphocyte count could be used for the diagnosis of CeD when other findings are not conclusive. In addition, the mentioned cells are also increased in treated CeD patients, which allow for diagnostic confirmation in patients already having a gluten-free diet, a diet that every day has more adepts. The manuscript is of great interest in the field and the data are well-discussed in general. However, the article has some issues that should be properly addressed to improve it.

Major Comments

-The search strategy seems very limited. Only the words celiac/coeliac were included. Some studies labeled as “gluten sensitive enteropathy” or similar will be missed. In addition, the language limitation to English is not ideal. Some studies from Europe or South America will be missed. Limitation to one language is not recommended in MA, and this should be described as a limitation of the review.

-The numbers don’t add up in figure 1 prisma. Revise.

-Selection process is unclear. Although well explained in the text, addition of new studies after revision of bibliography of selected papers should be included.

-Reason for study exclusion is unclear.

-What do the authors consider as significant heterogeneity? Are reasons for heterogeneity (subgroup analyses etc..) explored when significantly high; ie for TCR gamma delta +IEL? Authors mention the hypothesis for potential reasons but did not perform subgroup analyses to explore this. The main papers selected and reaching the criteria for final analysis are from the same country (Spain 5 out 6). Could there be a geographic bias? This could be derived of the language limitation selection? Do authors think that similar patterns would be found in other European countries or North-America? Although the issue is mentioned in the discussion and partially explained as intraepithelial lymphogram technique being widely utilized in the country and included in the Ministry of Health guidelines for the early diagnosis of CD, the subject has to be better discussed.

-The manuscript misses recent and relevant quotes describing characteristics of gamma delta CD3+ IELs. For instance, Mayassi et al.  “Chronic Iflammation Permanently reshapes tissue-residnet immunity in CeD”. Cell 2019.  Although I could understand exclusion of the paper because it did not meet the criteria (lack of CD3- determination) of the meta-analysis, I think it should be introduced and discussed at the very least.  

-I think the determination of CD3 negative IEL as inclusion criteria in the meta-analysis highly reduced the number of studies. Although I understand the reason why the authors decided to include it (specificity of the assay) and it is well explained in the manuscript, I think author would have also obtained interesting data looking at all the studies measuring CD3+ gamma delta by flow cytometer without other cell gathering restriction.

Minor Comments

Please provide figures with better quality (Figure 2-5).

Author Response

Thanks for your comments. We have dealt with as follows:

-The search strategy seems very limited. Only the words celiac/coeliac were included. Some studies labeled as “gluten sensitive enteropathy” or similar will be missed. In addition, the language limitation to English is not ideal. Some studies from Europe or South America will be missed. Limitation to one language is not recommended in MA, and this should be described as a limitation of the review.

Answer: As suggested by the referee we have added the term ‘gluten sensitive enteropathy’ to the search. No other paper fulfilled the inclusion criteria. As previously stated, the language limitation included English and Spanish (line 81). We also review the English abstracts of articles written in other languages, and no one fulfilled the inclusion criteria. This has been added to text (line 81).

-The numbers don’t add up in figure 1 prisma. Revise.

Answer: We have revised the numbers. We have added 15 articles with the new search. Full text articles assessed for eligibility were 49. From that, 44 were excluded, and 1 additional was obtained after reviewing the references of included papers. Thus, 6 articles were selected for the analyses.

-Selection process is unclear. Although well explained in the text, addition of new studies after revision of bibliography of selected papers should be included.

Answer: As previously stated ‘Bibliographies of all eligible studies that were identified were also comprehensively searched for studies not identified using the initial search strategy’ (line 91). In fact there was 1 article obtained after reviewing the references of selected articles, as is described in Figure 1 (Prisma selection process).

We have added a sentence to better explain the results of the selection process (line 127)

-Reason for study exclusion is unclear.

Answer: Main reasons for study exclusion were not fulfilling the inclusion criteria:

Articles allowing for the construction of a two-way table, extracting true positive, false positive, true negative and false negative values for TCRgammadelta+ IEL counting and/or coeliac lymphogram for the diagnosis of CD. Assessment of TCRgammadelta+ and CD3neg by flow cytometry. A clear description of how the diagnosis of CD had been performed.

This has now been added (lines 82 to 86) to the rest of exclussion criteria.

-What do the authors consider as significant heterogeneity? Are reasons for heterogeneity (subgroup analyses etc..) explored when significantly high; ie for TCR gamma delta +IEL? Authors mention the hypothesis for potential reasons but did not perform subgroup analyses to explore this. The main papers selected and reaching the criteria for final analysis are from the same country (Spain 5 out 6). Could there be a geographic bias? This could be derived of the language limitation selection? Do authors think that similar patterns would be found in other European countries or North-America? Although the issue is mentioned in the discussion and partially explained as intraepithelial lymphogram technique being widely utilized in the country and included in the Ministry of Health guidelines for the early diagnosis of CD, the subject has to be better discussed.

Answer: The Cochran-Q method and inconsistency index (I2) were adopted to investigate and quantify heterogeneity among the studies. Heterogeneity is to be expected in meta-analyses of diagnostic test accuracy (line 305) since the included studies used different cut-offs, and a lower cut-off increases sensitivity and decreases specificity, and vice versa. This was previously discussed (lines 299-306).

Meta-regression analyses taking into account age (paediatric versus adult populations) could not be performed due to the scarce number of articles included as was previously stated (line 264). The influence of age was commented in the discussion section (lines 262 to 276).

A sentence about the possibility of a geographical bias has been included (line 318-321): ‘Finally, since all included articles except one had been carried out in Spain it could be argued if the results could be extrapolated to other populations. However, the increase of TCRgd+ in patients with CD has been described in studies from other countries around the world [13,41-45]’.

-The manuscript misses recent and relevant quotes describing characteristics of gamma delta CD3+ IELs. For instance, Mayassi et al.  “Chronic Inflammation Permanently reshapes tissue-residnet immunity in CeD”. Cell 2019.  Although I could understand exclusion of the paper because it did not meet the criteria (lack of CD3- determination) of the meta-analysis, I think it should be introduced and discussed at the very least.  

Answer: Our paper was not focused in describing the role of TCRgammadelta+ cells, and thus, papers assessing their pathophysiological role were not included. As mentioned we only included articles allowing for the construction of a two-way table, extracting true positive, false positive, true negative and false negative values for TCRgammadelta+ IEL counting and/or coeliac lymphogram (TCRgammadelta+ and CD3- cells) for the diagnosis of CD. These are the articles in which these cells were measured for diagnostic purposes.

-I think the determination of CD3 negative IEL as inclusion criteria in the meta-analysis highly reduced the number of studies. Although I understand the reason why the authors decided to include it (specificity of the assay) and it is well explained in the manuscript, I think author would have also obtained interesting data looking at all the studies measuring CD3+ gamma delta by flow cytometer without other cell gathering restriction.

Answer: As we mentioned we included all articles using the TCRgammadelta+ count for diagnosis purposes, and also all the studies in which the celiac lymphogram (increase in TCRgammadelta+ plus decrease in CD3-) was assessed also for diagnosing purposes. In this sense, we performed a first meta-analysis for the measurement of the isolated increase of TCRgammadelta+ IEL, and a second meta-analyses for those studies evaluating the diagnosing accuracy for CD of the celiac lymphogram.

Minor Comments

Please provide figures with better quality (Figure 2-5).

Answer: These are the figures obtained from STATA software, and they are provided with the higher quality available.

Reviewer 2 Report

General: The authors have identified an interesting research question. “Systematic Review And Meta-Analysis: Accuracy Of Both Cd3 + T-Cell Receptor Gamma Delta+ Intraepithelial Lymphocyte Count And Coeliac Lymphogram Evaluated By Flow Cytometry For Coeliac Disease Diagnosis.” is an interesting topic.

The title is too large, please make it short and do not capitalize the words.

Abstract: look ok

Introduction:

1st paragraph: please use the latest data from the article “Global Prevalence of Celiac Disease: Systematic Review and Meta-analysis” to show the global prevalence of celiac disease. Text at https://www.cghjournal.org/article/S1542-3565(17)30783-8/fulltext

Lines 47-48 are unclear and needs to be rephrased or rewritten.

All abbreviations should be expanded on their first use throughout the manuscript. If using any gene names then please italicize them.

Methods:

You have used QUADAS-2 to assess the quality of the study. Please include a section t the end of methods paragraph to assess the “Confidence in cumulative evidence” if applicable.

Results section look ok.

Discussion Section:

Please include the official guidelines from World Gastroenterology Organisation Global Guidelines. Please include how these diagnostic tests are more useful when compared to their guidelines. They recommend second biopsies for “whom the first biopsies and serologic tests have been inconclusive “

Limitations need to be explained in more detail.

Please use the last section as a conclusion paragraph.

English and grammar needs to be thoroughly checked throughout the manuscript.

Tables and graphs are appropriate.

Overall a well conducted study.

Author Response

Thanks for your comments. We have dealt with them as follows:

The title is too large, please make it short and do not capitalize the words.

Answer: The initial title was: ‘Systematic Review And Meta-Analysis: Accuracy Of Both CD3+ T-Cell Receptor Gamma Delta+ Intraepithelial Lymphocyte Count And Coeliac Lymphogram Evaluated By Flow Cytometry For Coeliac Disease Diagnosis’.

We have shortened it to: ‘Systematic Review And Meta-Analysis: Accuracy Of Both Gamma Delta+ Intraepithelial Lymphocytes And Coeliac Lymphogram Evaluated By Flow Cytometry For Coeliac Disease Diagnosis’.

Introduction:

1st paragraph: please use the latest data from the article “Global Prevalence of Celiac Disease: Systematic Review and Meta-analysis” to show the global prevalence of celiac disease. Text at https://www.cghjournal.org/article/S1542-3565(17)30783-8/fulltext

Answer: We have changed reference number 1 according to reviewer suggestion.

Lines 47-48 are unclear and needs to be rephrased or rewritten.

Answer: This sentence has been rephrased.

All abbreviations should be expanded on their first use throughout the manuscript. If using any gene names then please italicize them.

Answer: Ok

Methods:

You have used QUADAS-2 to assess the quality of the study. Please include a section at the end of methods paragraph to assess the “Confidence in cumulative evidence” if applicable.

Answer: As the referee mentions we assessed the quality of the included studies using QUADAS-2, and also we use MIFlowCyt guidelines to assess the methodological quality of flow cytometry analysis. This was described in the Methods section (lines 102 to 110). These two tools describe the quality of cumulative evidence evaluated.

Discussion Section:

Please include the official guidelines from World Gastroenterology Organisation Global Guidelines. Please include how these diagnostic tests are more useful when compared to their guidelines. They recommend second biopsies for “whom the first biopsies and serologic tests have been inconclusive “

Answer: This has been added in the discussion section (lines 243 to 246).

Limitations need to be explained in more detail.

Answer: A comment on a possible geographical bias has been included.

Please use the last section as a conclusion paragraph.

Answer: Done

English and grammar needs to be thoroughly checked throughout the manuscript.

Answer: We have done our best with the English language, and the article was reviewed by a native speaker.

Round 2

Reviewer 1 Report

The author's have responded appropriately.